# Organic and Inorganic Selenium Compounds Affected Lipidomic Profile of Spleen of Lambs Fed with Diets Enriched in Carnosic Acid and Fish Oil

**DOI:** 10.3390/ani14010133

**Published:** 2023-12-29

**Authors:** Małgorzata Białek, Agnieszka Białek, Wiktoria Wojtak, Marian Czauderna

**Affiliations:** 1The Kielanowski Institute of Animal Physiology and Nutrition, Polish Academy of Sciences, 05-110 Jabłonna, Poland; m.bialek@ifzz.pl (M.B.); a.bialek@ifzz.pl (A.B.); w.wojtak@ifzz.pl (W.W.); 2School of Health and Medical Sciences, University of Economics and Human Sciences in Warsaw, Okopowa 59, 01-043 Warsaw, Poland

**Keywords:** selenium, carnosic acid, ovine spleen, fatty acids, tocopherols

## Abstract

**Simple Summary:**

The spleen, traditionally associated with its role in immune surveillance and blood cell turnover, nowadays is known to be engaged in metabolic control processes, e.g., in the metabolism of lipids. While input of energy sources is essential during animal ratio’ formulation, exploring the lipid composition of the spleen and its potential modulation by antioxidant supplements becomes particularly relevant. Our results may be practically applied in the food industry, as they may provide animal food, ensuring the nutritional requirements of especially poorly nourished consumers. Moreover, our findings could bridge the existing knowledge gap about the interplay of diet and lipid composition in the spleen. As this organ is considered to have an essential role in the development of atherosclerosis, obesity, nonalcoholic liver disease, nonalcoholic steatohepatitis, and fatty liver, understanding the function of this internal organ may be a starting point for developing efficient prevention strategies in order to counteract these disorders.

**Abstract:**

The purpose of our study was to investigate the effect of 0.35 mg Se/kg basal diet (BD) (Se as sodium selenate (Se^6^) and yeast rich in seleno-methionine (Se^Ye^)) and 0.1% carnosic acid (CA) supplementation to the diet containing 1% fish oil (F-O) and 2% rapeseed oil (R-O) on the contents of fatty acids (FA), malondialdehyde (MDA), tocopherols (Ts), and total cholesterol (TCh) in lambs’ spleens. A total of 24 male lambs (4 groups per 6 animals) have been fed: the control diet—the basal diet (BD) enriched in F-O and R-O; the CA diet—BD enriched in F-O, R-O, and CA; the Se^Ye^CA diet—BD enriched in F-O, R-O, CA, and Se^Ye^; the Se^6^CA diet—BD enriched in F-O, R-O, CA, and Se^6^. Dietary modifications affected the profiles of saturated (SFA), monounsaturated (MUFA), and polyunsaturated (PUFA) fatty acids in spleens. The Se^Ye^CA and Se^6^CA diets increased the docosapentaenoic acid preference in Δ4-desaturase; hence, a higher content of docosahexaenoic acid was found in the spleens of Se^Ye^- or Se^6^-treated lambs than in spleens of animals receiving the CA and control diets. The Se^Ye^CA and Se^6^CA diets increased the concentration ratio of n-3long-chain PUFA (n-3LPUFA) to FA (n-3LPUFA/FA) in spleens compared to the control and CA diets. The content of n-3PUFA was higher in the spleens of Se^6^ treated lambs than in spleens of animals receiving the Se^Ye^CA, CA, and control diets. The Se^6^CA diet increased the content of c9t11CLA in the spleen compared to the control, CA, and Se^Ye^CA diets. Experimental diets reduced the level of atherogenic FA, the content ratios of n-6PUFA/n-3PUFA and n-6LPUFA/n-3LPUFA, and improved the content ratio of MUFA/FA and the value of the hypocholesterolemic/hypercholesterolemic FA ratio in the spleen in comparison with the control diet. The experimental diets supplemented with Se^Ye^ or Se^6^ increased levels of TCh and Ts in spleens in comparison with the CA and control CA diets. The present studies documented that Se^6^, Se^Ye^, and CA influenced the metabolism of FA, Ts, and cholesterol in spleens.

## 1. Introduction

The spleen, derived from mesenchymal tissue, is the largest lymphatic organ found in all vertebrates [1]. The spleen (the multipurpose internal organ) has very important physiological roles in regard to red blood cells, storage of blood, the centre of the blood defence system, the immune system, as well as the formation of lymphocytes and eliminating senescent erythrocyte cells [2,3,4]. Moreover, the spleen controls physiologically essential processes, e.g., metabolism of metals, albuminoids, and, what is recently gaining increasing attention, the metabolism of lipids. The latter encompasses digestion and absorption, transportation through the blood, as well as biosynthesis [5]. There are several mechanisms presumably implicated in the spleen regulation of lipid metabolism. The most well-known is the theory of the splenic lipid reservoir, which covers both the volume of the spleen and the activity of macrophages. Hence, the presence of a mononuclear system of phagocytes active against fractions of lipids, the biosynthesis of anti-oxLDL antibodies together with the removal of antigen-antibody species, interference with the lipid peroxidation in the liver (liver-spleen axis), activity of lipoprotein lipase, shifts in expression of microRNAs involved in regulation of genes correlated with high density lipoprotein (HDL) metabolism, the platelet pathway, and immune-mediated mechanisms are also involved in lipid metabolism regulation [5]. Additionally, a splenic connection with the propagation of certain diseases connected with lipid disorders (like Niemann-Pick’s disease, Gaucher’s disease, Fabry disease or gangliosidoses) was also reported [6]. Total splenectomy may unfavorably affect levels of plasma lipids (triglycerides, cholesterols, fatty acids (FA)), and lipoproteins [7,8] and thus lead to the development of atherosclerosis and other cardiovascular disorders. The influence of the spleen in lipid metabolism was confirmed in rat, rabbit, and dog models [9,10,11]. However, to the authors’ best knowledge, no research was undertaken in the ruminants’ model. That is why our current experiment, aiming for the evaluation of the lipidomic profile of an ovine spleen, seems to be up-to-date and valid.

Determinations of contents of FA (especially polyunsaturated fatty acids (PUFA)) in ovine spleen were previously performed, e.g., to confirm the possible routes of supply of FA to the lymph. The most abundant lipid classes were phospholipids (phosphatidylethanolamine, phosphatidylcholine), free cholesterol, and triacylglycerols. In each lipid class, the amounts of the essential fatty acids (EFA) were lower than in the corresponding lipids of plasma or lymph [12]. Lymph nodes and the spleen jointly make up the majority of peripheral immune tissues, and earlier investigations have documented that spleen tissue is very sensitive to changes in concentrations of selenium (Se) in diets [4]. In fact, Se (the well-known antioxidant) is the essential part of glutathione peroxidases (GPx), which selenoenzymes function as stimulators of the immune system, responsible for scavenging free radicals, thus reducing oxidative damage (especially the decomposition of PUFA) in animal tissues [13]. In fact, dietary deficiency of Se stimulated inflammation and oxidative stress in the spleen, thereby disturbing the immune activity of the spleen [4,14]. Low levels of Se in the diet reduced the expression of seleno-proteins and Se contents, obstructed the thioredoxin and glutathione antioxidant systems, and caused a disturbance of the redox balance in the spleen. Too low Se contents in tissues stimulated the HIF-1α and NF-κB transcription factors, increasing pro-inflammatory cytokines (like IL-1β, IL-6, IL-8, IL-17, or TNF-α), reducing anti-inflammatory cytokines (e.g., IL-10, IL-13, or TGF-β), and stimulating expression of the downstream genes iNOS and COX-2, thus causing inflammation [4,14,15]. Additionally, insufficient doses of Se stimulated apoptosis (via the mitochondrial apoptotic pathway), up-regulated apoptotic genes, and down-regulated anti-apoptotic genes (*Bcl-2*) (at the mRNA level). On the other hand, diets containing high concentrations of Se and PUFA (particularly long-chain PUFA (LPUFA)) stimulated oxidative stress (i.e., oxidative damage) as well as reduced immune responses in splenocytes [16,17,18,19,20,21]. Therefore, optimal concentrations of PUFA and especially antioxidants (like Se compounds) in diets are very important for the proper function of the spleen [22,23,24,25,26,27,28,29]. In fact, oxidative stress can cause spleen disturbances (i.e., deteriorate immunological, hematological, or scintigraphic parameters used for assessing splenic functions) [22].

Recent studies have shown that diets supplemented with n-3PUFA, especially n-3LPUFA (derived from fish oil (F-O)), suppressed pro-inflammatory cytokine production, increased the number of lymphocyte cells, and stimulated the formation of immunocompetent cytokines in the spleen, which have a crucial role in anti-tumour and anti-infection activities [19,20,21,23]. Therefore, it is reasonable to add F-O rich in n-3LPUFA to an ovine diet. Importantly, PUFA, particularly LPUFA, derived from dietary F-O and rapeseed oil (rich in linolenic acid; LA), significantly reduced the capacity of the ruminal biohydrogenation (BH) of unsaturated fatty acids (UFA) and stimulated bacterial isomerization of UFA in the rumen [28]. As a consequence, dietary F-O and rapeseed oil (R-O) increased ruminal concentrations of PUFA and particularly health-promoting conjugated FA (i.e., products of bacterial isomerization of LA derived mainly from R-O) [28]. However, higher contents of UFA and LPUFA (especially n-6LPUFA) in internal organs and tissues stimulated oxidative stress in animals’ bodies [23,28]. Oxidative stress caused by reactive nitrogen (RNS) and oxygen (ROS) species damages lipids and proteins, as well as cellular RNA and DNA. Carbonyl compounds, especially malondialdehyde (MDA), the naturally occurring byproducts of PUFA peroxidation and prostaglandin synthesis, are known to be detrimental for health [24,28]. Taking into account that the principal physiological functions of more than half of Se-enzymes (e.g., GPx, selenoprotein P, or thioredoxin reductase) are to maintain the proper oxidative–antioxidant balance and low contents of ROS, RNS, and free radicals within cells [25], an adequate amount of Se should be delivered in diets. However, not only the quantity but also the chemical form of Se are of utmost importance. Organoselenium compounds (particularly seleno-methionine (Se-Met) derived from yeast rich in Se-Met (Se^Ye^)) are more efficiently incorporated in mammalian organisms than inorganic seleno compounds (like sodium selenite (Se^4^) or sodium selenate (Se^6^)) [24]. Importantly, Se-Met derived from Se^Ye^ is metabolized to inorganic seleno compounds or efficiently accumulated into ruminal microbiota and animal tissue proteins as Se-Met (as a replacement of methionine (Met)) or seleno-cysteine [26].

Considering the different metabolisms of Se compounds and thus their physiological role, which cannot be limited only to antioxidant properties, we claim that an additional antioxidant should be added to diets enriched in F-O (rich in n-3LPUFA) and R-O (rich in n-6PUFA, particularly LA). The herbal nutraceutical, carnosic acid (CA), which is one of the diterpenes present in rosemary [27], was chosen in this study. As it has been found previously, CA introduced in the ruminant ration improved production parameters (like diet intake, feed conversion efficiency, or live weight gain) and the growth and/or activity of rumen microorganisms [28,29,30]. Importantly, dietary CA modifies ruminal microbiota, the capacity of isomerization of UFA and BH of UFA, and, hence, the ruminal biosynthesis of VFA (in a dose-dependent manner) and profiles of FA and health-promoting conjugated FA in ruminants’ tissues and internal organs of farm animals [28,29,30,31,32,33]. CA (like Se compounds) possesses antioxidative properties and hence protects PUFA, particularly LPUFA, derived from dietary R-O and F-O, from peroxidation damage in ovine tissues [28,32,34,35,36,37,38,39]. Furthermore, our previous studies showed that diets enriched in Se (as Se^Ye^ or Se^6^), CA, and F-O affected the biosynthesis yield of Se-proteins as well as the profile of lipid compounds in the rumen microbiota [26,31,32], blood [28], muscles [35], adipose tissues [34,35], and internal organs like the brain [36], kidneys [37], heart [38], and pancreas [39] of farm animals (like lambs). That is why we also hypothesized that the bioaccumulation of FA, total cholesterol (TCh), tocopherols (Ts) (antioxidants), and the MDA content in the ovine spleen depend upon Se compounds (antioxidants) added to the diet enriched in F-O, R-O, and CA. Importantly, recent studies documented that Se^4,^ Se^6^ or Se^Ye^ added to diets enriched in F-O, R-O, or sunflower oil stimulated the accumulation of PUFA (like conjugated FA, LA, C20:5n-3, C22:6n-3, or C20:4n-6) in some tissues of lambs and rats [35,40,41]. In fact, our previous studies documented that dietary Se^Ye^ or Se^6^ stimulated the capacity of fatty acid elongases and desaturates as the concentrations of PUFA (especially LPUFA) increased in the livers, muscles (*musculus longissimus dorsi* and *musculus biceps femoris)*, and hearts of lambs when compared to the control animals [35]. 

Therefore, the principal objectives of our study were to evaluate the effect of Se^Ye^ (yeast rich in Se-Met) and Se^6^ (Na_2_SeO_4_) on the lipidomic profile, the concentration of Ts, and the yield of lipid peroxidation (i.e., the MDA level) in the ovine spleen. It is of importance not only from the point of view of ruminant physiology and particularly the welfare of lambs with diminished splenic functions but also human nutrition, especially in undeveloped countries. Indeed, the spleen, classified as giblets, may be considered an inexpensive source of bioactive lipids for humans at risk of malnutrition. However, animal products containing oxidized lipids, oxidized forms of Chol, and highly toxic MDA are harmful to consumers’ health. Obviously, examining the impact of Se^Ye^ and Se^6^ added to the ovine diet on the health-promoting properties of spleen is the secondary goal of our current studies. 

## 2. Materials and Methods

### 2.1. Lambs, Housing, Experimental Scheme, Diets and Sampling

Our research was accepted by the Third Local Commission of Animal Experiment Ethics—approval number: 41/2013 (the Warsaw University of Life Sciences; 8 Ciszewskiego street; Warsaw 02-786; Poland). Our manuscript does not contain human studies. All experiments were carried out on 24 Corriedale male lambs (the initial average body weight (BW) of 23.3 ± 2.1 kg) selected from 110 animals, according to their BW and age (82–90 days). Preliminary feeding and all nutritional experiments on lambs and then spleen collections were conducted in special animal laboratory rooms at the Kielanowski Institute of Animal Physiology and Nutrition, Jabłonna (Polish Academy of Sciences; Warsaw, Poland). Welfare guidelines for animals were strictly adhered to throughout the 3 weeks of preliminary and whole experimental period. Selected sheep were divided into four equinumerous groups of six lambs each, housed in a climate-controlled and ventilated facility (20–22 °C), and freely given access to tap water throughout the whole experiment. All animals were housed singly in adjacent pens (the height, length, and width of the pens were respectively 150, 170, and 130 cm) and had visual contact with other lambs. During the 3 weeks of the preliminary period, animals had free and *ad libitum* access to a basal diet (BD) supplemented with the vitamin and mineral premix (20 g in 1 kg of the BD), R-O (20 g R-O in 1 kg of the BD), and odorless F-O (10 g F-O in 1 kg of the BD) (Table 1). The BD contains: meadow hay (~36.0%), barley meal (~16.5%), soybean meal (~36.0%), wheat starch (~9%), and a mixture of vitamins and minerals (the premix—ID number: PL 1 405 002 p). The control and all experimental diets were iso-proteinous and iso-energetic. Details of the chemical composition of all ingredients in the BD are presented in Table 1. 

After 3 weeks of preliminary period, 5 weeks of experimental period were carried out during which animals received the BD containing 1% F-O and 2% R-O (the control diet) or 3 experimental diets enriched in 1% F-O, 2% R-O, and antioxidant(s) (i.e., 0.1% CA without/with 0.35 ppm selenium as Se^Ye^ or Se^6^). The control and all experimental diets (Table 2) were supplied to lambs at 7:30 am and 4:00 pm (in equal amounts). The amount of the diets was adjusted each week according to the nutritional requirements of lambs and their BW to avoid refusal of the offered diets. The average feed intake was 1.08 kg diet/lamb/day during the whole experimental period. Thus, each animal ate 37.8 kg of the experimental diet or the control diet. After 5 weeks of experimental period, ewes were rendered unconscious by intramuscular injections of 2–4 mg xylazine/10 kg of BW, and then lambs were immediately slaughtered (in accordance with the guidelines No. 1099/2009 of the European Union Council Regulations (EC)). Then the spleen was removed from each animal. All collected spleens were individually homogenized and finally stored at 32 °C (in tightly sealed vials).

### 2.2. Reagents, Chemicals and Dietary Supplements

GC-grade n-hexane (≥99%), GC-grade methanol (≥99.9%), and acetonitrile (HPLC-grade; ≥99.9%) were purchased from Lab-Scan (Dublin, Ireland). Isomers of conjugated linoleic acid (C18:2 isomers; CLA isomers), sorbic acid (as the internal standard; C6:2), nonadecanoic acid (as the internal standard; C19:0), and a mixture of 37 methylated fatty acid standards, α-tocopheryl acetate, α-, δ-, and γ- forms of tocopherol, 2,6-ditert-butyl-pcresol, sodium selenate (Se^6^), cholesterol, 25% aqueous 1,5-pentanedialdehyde solution, 1,1,3,3-tetramethoxy-propane (99%), 2,4-dinitrophenylhydrazine (including ~30% water), the methanolic solution of 25% BF_3_, and trichloroacetic acid were obtained from Sigma Aldrich (St Louis, MO, USA). NaOH, KOH, CHCl_3_, CH_2_Cl_2_, Na_2_SO_4_, and NaCl were obtained from Avantor Performance Materials (Gliwice, Poland). Other chemical reagents were of analytical grade. Water was applied for the preparation of all reagents was obtained from an Elix^TM^ water purification system (Millipore, Etobicoke, ON, Canada).

CA was obtained from Hunan Geneham Biomedical Technology Ltd. (Changsha Road, Changsha, China). The vitamin and mineral premix (ID No. a PL 1 405 002 p) was supplied from POLFAMIX OK (Trouw Nutrition, Grodzisk Mazowiecki, Poland). Se^Ye^ (highly selenized *Saccharomyces cerevisiae* yeast) was purchased from Sel-Plex (Alltech In.; Nicholasville, KY, USA); approx. 83% of the total amount of Se in Se^Ye^ is in the chemical form of Se-Met, while approx. 5% of Se is in the chemical form of Se-Cys; these Se-amino acids are in the proteins of yeast. R-O and odourless F-O, purchased from Company AGSOL (Pacanów; Poland), were stored at ~4 °C in a dark place in tightly closed bags. 

The energy content of F-O and R-O was 36.8 and 37.0 MJ/kg oil, respectively. Odourless F-O contained the following main FA (mg/kg F-O): C12:0 82, C14:0 12,345, *c9*C14:1 215, C15:0 477, C16:0 56,947, *c7*C16:1 318, c9C16:1 420, ΣC16:2 15,586, C17:0 493, *c9*C17:1 193, C18:0 9 452, *c6*C18:1 188, *c7*C18:1 842, *c9*C18:1 290,592, *c12*C18:1 15,834, *c14*C18:1 159, *c9c12*C18:2 (LA) 114,512, *c9c12c15*C18:2 (αLNA) 20,968, *c11*C20:1 24,206, *c7c9c12c15*C18:4 473, *c11c14*C20:2 2 270, *c8c11c14*C20:3 258, *c5c8c11c14*C20:4 (AA) 304, *c8c11c14c17*C20:4 607, C22:0 139, *c13*C22:1 11,036, *c11*C22:1 1 704, *c5c8c11c14c17*C20:5 (EPA) 6792, *c13**c16*C22:2 95, *c7c10c13c16*C22:4 144, *c15*C24:1 397, *c7c10c13c16c19*C22:5 (DPA) 1560, and *c4c7c10c13c16c19*C22:5 (DHA) 26,570. R-O comprised the following main FA (mg/kg R-O): C14:0 56, C16:0 13,091, *c9*C16:1 33, C18:0 5 490, *c9*C18:1 385,859, *c12*C18:1 786, LA 282,394, αLNA 38,474, C20:0 194, *c11*C20:1 108, C22:0 430, and *c15*C24:1 61.

### 2.3. Pre-Column Methods and Chromatography Instruments

#### 2.3.1. Fatty Acid Extraction and Methylation of Fatty Acids

The finely homogenized spleens (45–60 mg) from each lamb were hydrolyzed according to our previous published method [33]. Nonadecanoic acid (C19:0) was added to each processed spleen. Next, mild acid- and base-catalysed esterefications were used for the synthesis of fatty acid methyl esters (FAME) in analysed spleens. Next, methylated FA in assayed spleens were analysed using capillary gas chromatography and mass spectrometry (GC-MS) [33]. GC-MS analyses were performed using a Shimadzu GCMS-QP2010 Plus EI, a BPX70 fused silica column (120 m × 0.25 mm i.d. × 0.25 mm film thickness), and a mass detector (Model 5973 N). FA (as FAME) identification in spleens was validated using the electron impact ionization spectra and compared to the retention times of FAME standards as well as the reference mass spectra library (National Institute of Standard and Technology NIST, 2007). The determination of FAME contents in analytical samples was based on total ion current (TIC mode) chromatograms or/and selected ion monitoring (SIM mode) chromatograms [33].

#### 2.3.2. Lipid Quality Indices

Atherogenic (_index_A^SFA^) [42], modified atherogenic (_index_A^SFA^/Toc) [39], and thrombogenic (_index_T^SFA^) [42] indices were calculated according to the FA concentrations using the following formulae:_index_A^SFA^ = (C16:0 + 4 × C14:0 + C12:0)/(Σn-3PUFA + Σn-6PUFA + ΣMUFA)
_index_A^SFA/Toc^ = _index_A^SFA^/(0.05 × CδT + 0.15 × CγT + 1.36 × CαTAc + 1.49 × CαT)
_index_T^SFA^ = (C18:0 + C16:0 + C14:0)/[(3 × Σn-3PUFA + 0.5 × Σn-6PUFA + 0.5 × ΣMUFA)/Σn-6PUFA)
where MUFA—monounsaturated fatty acids; CδT, CγT, CαTAc, and CαT—contents of δ-tocopherol, γ-tocopherol, α-tocopheryl acetate, and α-tocopherol, respectively; 0.05, 0.15, 1.36, and 1.49 are biological activity coefficients of assayed tocopherols [43]. The concentration sums of atherogenic (A-SFA) and thrombogenic (T-SFA) saturated fatty acids (SFA) were calculated using the following formulae:A-SFA = C16:0 + C14:0 + C12:0
T-SFA = C18:0 + C16:0 + C14:0

The spleen index was calculated as spleen weight (g)/lamb weight (kg) [4,44]. The ratio of hypocholesterolemic/hypercholesterolemic fatty acids (h/HCh ratio) was calculated using the following equation [45]:h/HCh ratio = (*c13*C22:1 + *c11*C20:1 + *c14*C18:1 + *c12*C18:1 + *c9*C18:1 + *c7*C18:1 + DPA + *c7c10c13c16*C22:4 + EPA + *c11c14*C20:2 + AA + *c6c9c12*C18:3 + ΣLNA + LA)/(C16:0 + C14:0)

#### 2.3.3. Chromatographic Analysis of TCh, α-TAc, Tocopherols and MDA in Spleens

Total cholesterol (TCh), α-tocopheryl acetate (α-TAc), α-tocopherol (α-T), δ-tocopherol (δ-T), and γ-tocopherol (γ-T) were analysed in spleen samples (60–80 mg) using a liquid chromatograph (UFLC-DAD; Shimadzu, Tokyo, Japan), including two LC-pumps (LC-20ADXP), an autosampler (SIL-20ACXR), a communications bus module (CBM-20A), a column oven (CTO-20A), a degasser (DGU-20A5), a SPD-photodiode array detector, and a Kinetex C18-column (Phenomenex; the particle size: 2.6 μm; Hydro-RP, 100 Å, 150 mm × 2.1 mm i.d.; Torrance, CA, USA) [46].

Concentrations of malondialdehyde (MDA) in ovine spleens (50–70 mg) were chromatographically quantified after pre-column saponification, followed by derivatization [33]. Concentrations of derivatized MDA in assayed spleens were determined applying a liquid chromatograph (UFLC-DAD; Shimadzu, Tokyo, Japan) equipped with a Synergi C18-column (Phenomenex; particle size: 2.5 µm; Hydro-RP, 100 Å, 100 mm × 2 mm i.d.; Torrance, CA, USA).

### 2.4. Statistical Analyses

Statistical analyses were carried out applying the Statistica 12.5 PL software package (StatSoft Inc., Tulsa, OK, USA). The results are presented as means and standard errors of means, except for the live weight (LW) of animals, feed conversion efficiency, and spleen weight (mean ± standard deviation). The Shapiro-Wilk test was applied to analyse the normality of the data distribution. The impact of the experimental diets on the examined parameters in ovine spleens for variables with normal distribution was tested applying one-way ANOVA and Tukey’s Honestly Significant Difference test. All results for variables without normal distribution were tested applying the Kruskal-Wallis test, which is the non-parametric equivalent of one-way ANOVA, with a post-hoc multiple comparison test. The acceptable level of statistical significance was established at *p* ≤ 0.05.

## 3. Results

Our observation showed no damaging symptoms (e.g., diarrhea and vomiting) in the control and all experimental lambs, as well as no visual pathological changes, acute toxicity, or toxic changes of Se (as Se^Ye^ or Se^6^) in the spleen and other organs (like the liver, kidneys, pancreas, and brain), muscles, and adipose tissues. Se^6^ and CA added to the ovine diet significantly increased the FCE, final LW, and BWG of sheep in comparison to the CA and Se^Ye^CA diets (Table 2). On the other hand, the Se^Ye^CA diet most efficiently elevated spleen weight and the values of spleen index in comparison with the Se^6^CA, CA, and control diets. 

### 3.1. Contents of SFA and MUFA in Ovine Spleens

Results reflecting the concentration of selected SFA in the spleen of sheep are presented in Table 3. Compared to the spleens of the control lambs (the control group), all experimental diets substantially reduced the C17:0 and C18:0 contents in the ovine spleens. Similarly, the experimental diets with CA, irrespective of the presence of Se (as Se^Ye^ or Se^6^), tend (*p* < 0.10) to decrease the C16:0 content in the spleen in comparison with the control diet, whereas the Se^Ye^CA and CA diets decreased (*p* ≤ 0.05) in the contents of C22:0 and C24:0 in the spleen compared to the Se^6^CA and control diets. The Se^6^CA and CA diets decreased (*p* ≤ 0.05) the content of A-SFA and did not influence (*p* > 0.05) T-SFA, the concentration sum of all assayed FA (ΣFA) and all assayed SFA (ΣSFA) in the ovine spleens as compared with the control diet. Compared to the control diet, the Se^6^CA diet decreased the ratios of T-SFA/ΣFA (*p* ≤ 0.05), ΣSFA/ΣUFA (*p* ≤ 0.05), and ΣSFA/ΣMUFA (*p* ≤ 0.05) and did not exert any impact on the ratios (*p* > 0.05) of A-SFA/ΣFA, ΣSFA/ΣPUFA, and ΣSFA/ΣFA in the ovine spleens. Moreover, the Se^6^CA diet most efficiently reduced the values of indexASFA and indexTSFA in the spleen in comparison with the Se^Ye^CA, CA, and control diets. The experimental diet containing only CA also significantly elevated the content ratios (*p* ≤ 0.05) of ΣSFA/ΣPUFA or did not influence the ratios (*p* > 0.05) of ΣSFA/ΣUFA and ΣSFA/ΣFA in the spleen when compared with the Se^Ye^CA, Se^6^CA, and control diets.

Results concerning the contents of MUFA in the ovine spleen are presented in Table 4. The CA diet significantly reduced (*p* ≤ 0.05) the amounts of *c9*C14:1, *c9*C16:1, *c12*C18:1, and *t11*C18:1 in the spleen as compared with the Se^Ye^CA, Se^6^CA, and control diets. The Se^6^CA diet substantially increased (*p* ≤ 0.05) the indices of the ∆9-desaturation of C16:0, *t11*C18:1, the total ∆9-desaturation (Σ∆9index) of C18:0, C16:0, C14:0, and t11C18:1, and the total ∆9-, ∆6-, ∆5-, and ∆4-desaturation (Σ^∆9,6,5,4^FAindex) of FA in the spleen when compared with the control diet (Table 4). Similarly, the Se^Ye^CA, Se^6^CA, and CA diets increased the values of Σ^∆9,6,5,4^FAindex and the content ratio of ΣMUFA/ΣFA in the spleen in comparison with the control diet (Table 4).

### 3.2. PUFA Concentrations in the Ovine Spleens

The obtained results showed (Table 5) that the Se^6^CA diet increased the contents of *c9t11*CLA, α-linolenic acid (αLNA), docosahexaenoic acid (DHA) and the concentration sum of all assayed n-3PUFA (Σn-3PUFA) in the ovine spleens as compared to the CA and control diets (*p* ≤ 0.05). The experimental diets with Se^Ye^ or Se^6^ increased (*p* ≤ 0.05) the values of the elongase index of C20 (^n−3ElongC22/C20^index) and ∆4-desaturation of DPA index (∆4_index_) in the spleen in comparison with the spleen of the lambs fed the control diet. In contrast, the experimental diet with only CA significantly reduced (*p* ≤ 0.05) the indices of the C18:0 elongase and ∆4-desaturation of docosapentaenoic acid (DPA) as well as the levels of LA, *c11c14*C20:2, αLNA, *c8c11c14*C20:3, arachidonic acid (AA), eicosapentaenoic acid (EPA), Σn-3PUFA, Σn-6PUFA, ΣPUFA, and Σn-6LPUFA in the ovine spleens as compared with the control diet. Similarly, the Se^Ye^CA diet significantly decreased (*p* ≤ 0.05) the levels of LA, αLNA, *c11c14*C20:2, AA, EPA, Σn-6PUFA, and Σn-6LPUFA in the ovine spleen in comparison with the control diet. Compared to the CA and control diets, lower concentration ratios (*p* ≤ 0.05) of Σn-6PUFA/Σn-3PUFA and Σn-6LPUFA/Σn-3LPUFA were found in the spleen of ewes fed the Se^Ye^CA and Se^6^CA diets. The substantially higher ratio of Σn-3LPUFA/ΣFA and the index of ∆4-desaturation of DPA were found in the spleen of animals fed the Se^Ye^CA and Se^6^CA diets than fed the CA and control diets (*p* ≤ 0.05). The Se^6^CA diet increased the ratio of hypocholesterolemic/hypercholesterolemic fatty acids (h/H-Ch ratio) in the ovine spleens in comparison to other diets.

### 3.3. Concentrations of Tocopherols, TCh and MDA in the Ovine Spleens

CA added to the diet caused a significant (*p* ≤ 0.05) decrease in the TCh content in the ovine spleens as compared with the Se^Ye^CA, Se^6^CA, and control diets (Table 6). In contrast, the experimental diets supplemented with Se^6^ or Se^Ye^ significantly decreased the levels of α-T, the content sums of α-TAc and α-T (Σ (α-T + α-TAc)), as well as all detected tocopherols (Σall-Ts) when compared to the CA and control diets. Compared to the CA diet, the Se^6^CA and Se^Ye^CA diets decreased the modified atherogenic index (_index_A^SFA^/Toc) [39] in the ovine spleens. Compared to the control diet, all experimental diets decreased the _index_A^ΣSFA/ΣToc^ values in the spleen of sheep. The Se^6^CA diet significantly decreased (*p* ≤ 0.05) the values of the PUFA peroxidation index (^ΣPUFA^MDA_index_) in the spleen in comparison with the Se^Ye^CA, CA, and control diets. On the other hand, the CA diet increased (*p* ≤ 0.05) the ^PUFA^MDA_index_ value in the ovine spleens when compared with the control, Se^Ye^CA, and Se^6^CA and control diets.

## 4. Discussion

Our current and previous studies showed that the control diet containing F-O and R-O as well as all experimental diets including CA, F-O, and R-O without/with Se (as Se^6^ or Se^Ye^ [47,48,49,50]) did not affect negatively the total health conditions, welfare, and especially growth parameters of lambs [28,32,33,34,35,36,37,38,39]. Furthermore, the results of current investigations were confirmed by previous studies that documented that neither spleen injury symptoms nor toxic symptoms and macroscopic lesions of 10 g of F-O, 20 g of R-O, and 0.35 mg of Se (as Se^6^ or Se^Ye^) added to 1 kg of the BD were observed in lambs [28,35,36,39,46]. Importantly, dietary oils (like R-O and especially F-O), CA, and Se compounds (as Se^Ye^, Se^6^, or Se^4^ [48,49,50]) primarily affect the ruminal microorganisms of ruminants [28,32,42,46,47]. Microorganisms (especially bacteria) in the rumen metabolized (i.e., anabolized and/or catabolized) ingredients in diets [42,46,47,48,51]. Indeed, dietary Se compounds are efficiently metabolized by ruminal bacteria [46]. Importantly, ruminal bacteria catabolized excessive amounts of dietary selenium (>0.5 mg Se/kg diet). Similarly, bacteria are isomerized with UFA as well as the BH of UFA (particularly LPUFA, which has a detrimental impact on ruminal bacteria) [28]. In fact, Vargas et al. [47] documented that dietary supplementation with F-O or sunflower oil reduced the numbers of ruminal *Butyrivibrio* C18:0-producers and influenced the numbers of *Streptococcus bovis*, *Selenomonas ruminantium*, methanogens, and protozoa, but not the total number of bacteria in a rumen. Similarly, our previous studies showed that dietary supplementation with 1% F-O, regardless of the presence of CA, Se^6^, or Se^Ye^, changed the composition of ruminal microbiota and, therefore, FA metabolism, decreased the BH yield of C18-UFA, and stimulated the bacterial isomerization of UFA [28,32,33,34,35,36,37,38,39]. Thus, sheep performance can be improved in diets with the inclusion of 12.0 g F-O/kg DM, whereas higher contents of R-O and especially F-O in diets result in impairment of all performance variables of lambs [28,32,33,34,35,36,37,38,39,47].

However, on the other hand, a certain part of supplements and/or ingredients in diets, avoiding (“the salvation”) bacterial metabolism in the rumen, passes to further sections of the digestive tract of ruminants. Therefore, diets including up to 2 mg of Se per kilogram of BD would not be toxic for sheep and cows, whereas long-term dietary supplementation of higher doses (>5 mg of Se/kg BD) of Se^6^ or especially Se^4^ and selenides (Se^2−^) can be teratogenic as well as hepatotoxic in bodies of ruminants [24,48]. Indeed, these physiological effects of dietary Se were confirmed in the spleen, as well as previously in adipose tissues, muscles, and other internal organs (like kidneys, pancreas, heart, or brain) [28,32,33,36,37,39,46]. Furthermore, lack of detrimental health impact of used Se supplements in our studies can be due to the fact that Se^6^ and particularly organic Se compounds derived from Se^Ye^ are relatively less reactive and toxic in mammals as compared to Se^4^ and especially Se^2−^ (like sodium selenide) [24]. Moreover, Se-Met (the main Se compound in Se^Ye^) is the less physiologically active chemical form of Se, therefore, dietary supplementation with Se^Ye^ is considered a safe storage mode for Se [24].

The mammals’ spleen is a very important immune organ, possessing different immunocompetent cytokines that effectively stimulate anti-cancer as well as anti-infective functions. The higher spleen weight as well as the higher value of the spleen index observed in the lambs receiving the Se^Ye^CA diet may suggest that Se-Met (derived from Se^Ye^) is predominantly accumulated in spleen proteins instead of Met [24]. These Se-Met-containing proteins in the spleen have no impact on important biochemical reactions, particularly protein or enzyme biosynthesis [49]. Moreover, Se-Met-containing proteins in the spleen are less effective in the detoxification of RNS, ROS, or other radicals in comparison to Se-Cys-enzymes biosynthesized primarily from SeVI supplemented with the SeVICA diet [50]. As a consequence, compared to sheep receiving the Se^6^CA diet, Se-Cys-enzyme deficiency in the spleen of sheep receiving the Se^Ye^CA diet caused a redox imbalance. Therefore, the current research is consistent with studies by Yan et al. [44] showing that the value of the spleen index decreased significantly (*p* ≤ 0.05) with increasing dietary supplementation with superoxide dismutase (SOD).

Moreover, the present research showed that especially the Se^Ye^CA diet significantly stimulated TCh incorporation in the ovine spleen, which was also observed in pancreases and kidneys [37,38,39]. It is well established that elevated levels of cholesterol and atherogenic SFA (as well as atherogenic index) in tissues are associated with increased oxidative stress and LDL-cholesterol concentration, as well as activating the inflammation process [52,53]. Similarly, a high-cholesterol diet increased ROS generation and formation in mitochondria and decreased the level of glutathione (an efficient free radical scavenger and a key antioxidant) [53]. These were confirmed by our results obtained on lambs fed particularly with the Se^Ye^CA diet. Indeed, the level of MDA tended to decrease (*p* > 0.05) and the index values of ^ΣPUFA^MDA_index_ and _index_A^SFA^ in the spleen of lambs receiving the Se^Ye^CA diet were lower (*p* ≤ 0.05) than in the spleen of ewes receiving the Se^6^CA diet. The earlier results [37,39] obtained for the pancreas and kidneys also seem to support the above-mentioned assumptions. Thus, we argued that the higher levels of ROS, RNS, or other radicals in the spleen of the Se^Ye^CA-treated lambs may cause inflammation of this internal organ [20]. Therefore, we supposed that the levels of TNF-α, IL-1β, IL-6, IL-8, or IL-17 (the pro-inflammatory cytokines [4]) in the lambs’ spleen of the Se^Ye^CA group were higher than those in the Se^6^CA group. So, the higher spleen index observed in the Se^Ye^CA-treated lambs may suggest increased amounts of splenocytes and, thus, enhanced immunoreaction, or state similar to hypersplenism, when macrophages in the spleen contain a large amount of fat due to hyper-active phagocytosis [5].

Our current results indicate that diets enriched with extra Se compounds exert lipogenic effects. In fact, the Se^6^CA and Se^Ye^CA diets significantly stimulated or tend to stimulate the incorporation of TCh in the ovine spleens when compared to the CA and control diets, which was also observed in kidneys [37]. Indeed, cholesterol as well as seleno-proteins use isopentenyl pyrophosphate for Sec-tRNA and isoprenoid synthesis [54]. In contrast, compared to the Se^Ye^CA and Se^6^CA diets, the CA diet reduced the concentration of TCh in the spleen, kidneys, heart, subcutaneous fat, and fat located between the thigh muscles of sheep [34,37,38]. So, these results are consistent with previous studies [55] indicating that CA reduced the BWG and concentrations of triglyceride, TCh, and glucose in experimental animals. Really, CA reduces the nuclear level of SREBPs (i.e., sterol regulatory element-binding proteins) as well as downregulates their target genes, hence reducing the yield of the de novo-biosynthesis of cholesterol and fatty acids; dietary CA stimulates the degradation of mature SREBPs-form [56].

This study revealed that the Se^Ye^CA and Se^6^CA diets statistically and significantly increased the yield of Δ9-desaturation of C16:0 in the ovine spleen in comparison with the CA and control diets. So, the present research indicated that the Se^Ye^CA diet and particularly the Se^6^CA diet affected a substrate preference in the Δ9-desaturase in the ovine spleens. In fact, the Δ9-desaturase prefers acyl-CoA [57] with lengths of saturated fatty acids containing 16-carbons (i.e., formation of palmitoyl-CoA desaturase) in the spleen of animals receiving the Se^Ye^CA and Se^6^CA diets as compared with the CA and control diets. Additionally, the Se^Ye^CA and Se^6^CA diets increased the concentrations of α-T and stimulated DPA preference in the Δ4-desaturase; hence a higher level of DHA was found in the spleen of the Se^6^- or Se^Ye^-treated lambs than in animals fed the CA or control groups. Thus, our current study is in agreement with earlier research [58,59] in which the Δ4-desaturase capacity and DHA content were correlated with the level of Se-dependent hormones and enzymes, as well as tocopherol concentrations in animal tissues [60]. Indeed, tocopherols and Se compounds play essential roles in Δ4-, Δ5-, and Δ6-desaturations of UFA by being involved in the microsomal electron transport chain and in the peroxidase group of the desaturase complex [60].

In contrast, the CA diet decreased C14:0 and C16:0 (as substrates) preference in the Δ9-desaturation, whereas increased t11C18:1 preference in the Δ9-desaturation in the ovine spleens in comparison with the Se^Ye^CA and Se^6^CA diets. Thus, we claimed that, compared to C14:0 and C16:0, t11C18:1 shows greater affinity to Δ9-desaturase in the spleen of ewes receiving the CA diet. However, Se^Ye^ and particularly Se^6^ added to experimental diets with CA stimulated C14:0 and C16:0 preference in Δ9-desaturation in the spleen as well as the values of the Δ9-desaturase index in the body fat of sheep [34] as compared with the CA and control diets.

The present and our previous research showed that the Se^Ye^CA and Se^6^CA diets increased the accumulation of TCh and Σall-Ts in the spleen, heart, and subcutaneous fat [34,38] in comparison with the CA and control diets. So, we claimed that dietary SeY or SeVI spared of tocopherols as well as easy peroxidized long-chain highly UFA in lambs’ tissues. In fact, dietary Se^Ye^ and Se^6^ are utilised for the biosynthesis of Se-dependent antioxidant enzymes, which prevent the peroxidation of UFA (particularly highly unsaturated long-chain FA) in mammalian organisms [49,59,61]. Thus, Se enzymes decreased the content of free-radical-mediated peroxidation, and synergistically with tocopherols, regulated lipid peroxidation in mammalian tissues [62].

In this study, it was shown that all experimental diets decreased the spleen content of fatty acids responsible for atherogenesis [52]. Indeed, particularly the Se^6^CA diet, decreased the content of A-SFA, values of the modified atherogenic index (_index_A^SFA/Toc^) and _index_T^SFA^, whereas improved the value of the h/H-Ch ratio in the ovine spleen in comparison with the control diet.

## 5. Conclusions

The present studies indicated the modulatory effect of seleno compounds and CA supplemented with diets containing F-O on lipid compound metabolism as well as oxidative stress in the ovine spleen without adverse effects and the disruption in animal physiology. The experimental diets (particularly the Se^6^CA diet) increased the content of c9t11CLA while reducing concentrations of FA responsible for atherogenesis and improving the value of the hypocholesterolemic/hypercholesterolemic FA ratio in the spleen. Moreover, dietary Se^Ye^ and Se^6^ significantly stimulated the accumulation of tocopherols (particularly α-T and α-TAc) in the ovine spleen. Thus, especially the experimental diets enriched in Se^Ye^ or Se^6^ improve nutritional status of spleen, which may be considered edible giblets. That is why obtained results seem promising not only when animal welfare, physiology, and health are considered (the principal goals of our study), but also when human nutrition is taken into account (the secondary goal of our study).

## Figures and Tables

**Table 1 animals-14-00133-t001:** Chemical profile of ingredients (the concentrate-hay diet with a mineral and vitamin mixture ^1^) in basal diet and chemical composition of control and experimental diets (means, *n* = 3).

Chemical Composition of the BD Ingredients ^2,3^, % of Dry Matter (DM)				
Indices	Meadow Hay ^5^	Concentrate ^6^		
		Barley Meal	Soybean Meal	Wheat Starch
Dry matter, % BD	88.4	87.6	89.7	87.3
Crude protein	9.50	9.94	41.81	0.90
Crude fibre	27.29	2.87	4.34	−
Crude fat	3.40	2.50	2.25	0.09
Ash	4.85	1.84	6.16	0.12
Neutral detergent fibre	59.17	18.02	18.81	−
Acid detergent fibre	32.08	4.61	6.44	−
Acid detergent lignin	4.47	1.14	1.49	−
Se content, mg Se/kg	0.003	0.016	0.020	− ^4^
**Chemical composition of the BD**				
**Indices**		**Amount**		
Dry matter, g/kg BD		884.3		
Crude protein, g/kg DM		201.9		
Crude fibre, g/kg DM		118.6		
Crude fat ^7^, g/kg BD		21.7		
Total crude fat ^8^, g/kg BD		51.7		
Ash, g/kg DM		42.8		
Neutral detergent fibre, g/kg DM		310.5		
Acid detergent fibre, g/kg DM		146.3		
Acid detergent lignin, g/kg DM		23.3		
Gross energy ^9^, MJ/kg DM		17.9		
Se concentration, mg Se/kg BD		0.16		

^1^ One kg of the BD contained 20 g of the mineral and vitamin mixture (premix); one kg of the mineral and vitamin mixture comprised: g: Ca 285, P 16, Na 56, Fe 1 (as sulphate), Cu 0.5 (as sulphate), Mn 5.8 (as sulphate), Zn 7.5 (as sulphate); mg: Co 42 (as carbonate), I 10 (as iodate), Se 6 (as sodium selenite); IU: vit. A 500,000, vit. D_3_ 125,000, and vit. E 25,000 (as α-tocopherol). ^2^ The gross energy (MJ/kg of DM): meadow hay 17.1, barley meal 16.3, soyabean meal 17.8, wheat starch 16.7. ^3^ The contents of toxic elements in the BD: mg/kg: As 1.39 ± 0.03, Cd 0.068 ± 0.001, Sb 0.0155 ± 0.0015, and Pb 0.514 ± 0.003. ^4^ The Se content in wheat starch was below the limit of detection (LOD). ^5^ Main fatty acid (FA) content in the meadow hay: mg/kg: C8:0 83, C12:0 142, C14:0 239, *cis9*C15:1 (*c9*C15:1) 131, C16:0 4 034, *c9*C16:1 184, C18:0 459, *c9*C18:1 1 266, *c12*C18:1 72, *c9c12*C18:2 (LA) 13,100, *c9c12c15*C18:3 (αLNA) 4 178, C20:0 58, *c11*C20:1 74, C22:0 101, C24:0 69, *c15*C24:1 71. ^6^ Main FA content in concentrate: mg/kg: C14:0 104, C16:0 3 189, C18:0 1 425, *c9*C18:1 774, LA 29,163, αLNA 1 014. ^7^ Crude fat originating from BD (i.e., the meadow hay and concentrate). ^8^ Total crude fat originating from the BD and added oils (i.e., rapeseed oil (R-O) and fish oil (F-O)). ^9^ Total gross energy of the BD enriched in R-O and F-O without or with antioxidant(s) (i.e., CA, Se as yeast rich in seleno-methionine (Se^Ye^) or Se^6^ (Se as sodium selenate)).

**Table 2 animals-14-00133-t002:** The experimental scheme and the extra ingredients in the experimental and control diets, LW (the live weight, kg) of sheep, BWG (body weight gain, kg), spleen weight (g), spleen index (g/kg), and FCE (feed conversion efficiency, kg/kg) of lambs.

The Experimental Scheme		The Live Weight (LW)			Spleen Weight		FCE ^5^ kg/kg
Group/Diet	Ingredients Added to 1 kg of the Basal Diet (BD)	Initial LW kg ^1^	Final LW kg ^2^	BWG kg	G ^3^	Spleen Index ^4^	
						g/kg Final LW	
Control ^6^	20 g R-O and 10 g F-O	30.6 ± 2.4	37.7 ± 2.1 ^ab^	7.1 ± 0.4 ^ab^	75.8 ± 3.3 ^a^	2.01 ^a^	0.189 ^ab^
CA ^7^	20 g R-O, 10 g F-O, and 1 g CA	30.6 ± 2.6	37.2 ± 2.3 ^b^	6.6 ± 0.4 ^b^	75.5 ± 3.1 ^a^	2.03 ^a^	0.174 ^b^
Se^Ye^CA ^7^	20 g R-O, 10 g F-O, 1 g CA, and 0.35 mg Se as Se^Ye^	30.3 ± 2.7	36.8 ± 2.7 ^b^	6.5 ± 0.4 ^b^	88.6 ± 3.5 ^b^	2.41 ^b^	0.174 ^b^
Se^6^CA ^7^	20 g R-O, 10 g F-O, 1 g CA, and 0.35 mg Se as Se^6^	30.3 ± 3.0	38.5 ± 3.1 ^a^	8.2 ± 0.4 ^a^	72.0 ± 3.1 ^a^	1.87 ^a^	0.215 ^a^
		*p* = 0.63	*p* = 0.04	*p* = 0.03	*p* = 0.03	*p* = 0.02	*p* = 0.02

BWG = final LW—Initial LW. Means with different superscripts (a, b) within columns are significantly different at *p* ≤ 0.05. R-O—rapeseed oil. F-O—fish oil. CA—carnosic acid. Se^Ye^—yeast rich in seleno-methionine; Se^6^—sodium selenate (Na_2_SeO_4_). ^1^ The initial live weight of sheep (mean ± SD) after the preliminary period; for the whole preliminary period, animals received the control diet. ^2^ The average live weight of sheep (mean ± SD) receiving the diets for 5 weeks of experimentation. ^3^ The average weight of the ovine spleens. ^4^ The relative weight of the ovine spleen (g/kg) = spleen weight (g)/the final LW of sheep (kg) [4]. ^5^ FCE for 5 weeks of experimentation; FCE = [BWG, kg]/[diet intake, kg]. ^6^ The Se level in 1 kg of the control diet: 0.16 mg Se/kg. ^7^ Se levels in the CA, Se^Ye^CA, and Se^6^CA diets (mg Se/kg diet): 0.16, 0.51, and 0.51, respectively. The Se levels in 1 kg of meadow hay and soybean and barley meals were 0.003 mg, 0.020 mg, and 0.016 mg, respectively; the Se level in wheat starch was below the limit of detection.

**Table 3 animals-14-00133-t003:** The contents (µg/g spleen) of selected individual saturated fatty acids (SFA), content sums of all analysed SFA (ΣSFA) ^1^, all analysed FA (ΣFA), values of the atherogenic [42] (_index_A^SFA^) and thrombogenic index [42] (_index_T^SFA^), and the content ratios of ΣSFA to the content sums of UFA (ΣSFA/ΣUFA), PUFA (∑SFA/ΣPUFA), MUFA (ΣSFA/ΣMUFA), and ΣFA (ΣSFA/ΣFA) in the ovine spleens.

	Additive: Group/Diet:	-	CA	CA and Se^Ye^	CA and Se^6^	SEM	*p* Value
Item		Control	CA	Se^Ye^CA	Se^6^CA		
C10:0		0.9	0.5	1.0	0.9	0.3	0.41
C12:0		1.0	0.7	0.6	1.1	0.4	0.29
C14:0		56.1	48.2	61.6	57.3	1.9	0.37
C15:0		0.4	0.2	0.3	0.3	0.2	0.19
C16:0		4 452	3 796	3 968	3 566	97	0.09
C17:0		112 ^c^	49 ^a^	84 ^b^	85 ^b^	5	0.04
C18:0		6 119 ^b^	5 183 ^a^	5 284 ^a^	5 086 ^a^	49	0.04
C20:0		1.0	0.8	0.7	1.1	0.1	0.29
C22:0		16.4 ^c^	8.6 ^a^	14.1 ^b^	19.5 ^c^	0.3	0.03
C24:0		56.4 ^b^	39.2 ^a^	45.7 ^a^	60.7 ^b^	0.5	0.04
A-SFA		4 509 ^b^	3 845 ^a^	4 031 ^ab^	3 625 ^a^	98	0.04
A-SFA/ΣFA		0.2106	0.2171	0.2191	0.2000	0.0022	0.13
T-SFA		10 627	9 028	9 314	8 709	198	0.13
T-SFA/ΣFA		0.5009 ^b^	0.5099 ^c^	0.5062 ^bc^	0.4806 ^a^	0.0017	0.04
_index_A^SFA^		0.4585 ^b^	0.4700 ^c^	0.4762 ^c^	0.4179 ^a^	0.0007	0.04
_index_T^SFA^		1.0399 ^c^	0.9925 ^bc^	0.8737 ^b^	0.7852 ^a^	0.0010	0.03
ΣSFA		10 816	9 127	9 460	8 877	223	0.09
ΣFA		20 959	17 655	18 350	18 150	667	0.11
ΣSFA/ΣUFA		1.0660 ^bc^	1.0697 ^c^	1.0636 ^b^	0.9567 ^a^	0.0014	0.04
ΣSFA/ΣPUFA		2.3420 ^a^	2.6507 ^c^	2.5652 ^b^	2.2345 ^a^	0.0041	0.03
ΣSFA/ΣMUFA		1.8919 ^c^	1.7787 ^b^	1.8039 ^b^	1.6728 ^a^	0.0059	0.04
ΣSFA/ΣFA		0.5102	0.5156	0.5143	0.4898	0.0082	0.37

SEM = standard error of the mean. Means in rows sharing the different superscript letter (a, b, or c) are significantly different at *p* ≤ 0.05. ^1^ The content sum of saturated fatty acids (ΣSFA) = C8:0 + C10:0 + C11:0 + C12:0 + C13:0 + C14:0 + C15:0 + C16:0 + C17:0 + C18:0 + C20:0 + C22:0 + C24:0.

**Table 4 animals-14-00133-t004:** The contents (µg/g spleen) of selected individual monounsaturated FA (MUFA) and indices of ∆9-desaturation of FA and total FA desaturation in the ovine spleens.

	Additive: Group/Diet:	-	CA	CA and Se^Ye^	CA and Se^6^	SEM	*p* Value
Item		Control	CA	Se^Ye^CA	Se^6^CA		
*c*9C14:1		62.9 ^c^	33.9 ^a^	48.2 ^b^	67.3 ^c^	3	0.03
*c*9C16:1		115 ^a^	99 ^a^	143 ^b^	153 ^b^	7	0.04
*c10*C16:1		7.08	7.58	7.08	8.24	0.83	0.53
*t11*C18:1		263 ^c^	155 ^a^	210 ^b^	237 ^bc^	11	0.03
c9C18:1		4 698	4 317	4 151	4 137	159	0.32
c*12*C18:1		537 ^a^	471 ^a^	622 ^b^	639 ^b^	20	0.04
*c11*C20:1		33 ^a^	36 ^a^	62 ^b^	65 ^b^	5	0.03
ΣMUFA ^1^		5 716	5 130	5 243	5 305	51	0.37
ΣMUFA/ΣFA		0.274 ^a^	0.291 ^b^	0.287 ^b^	0.292 ^b^	0.003	0.03
^C18:0^∆9_index_ ^2^		0.436	0.453	0.441	0.449	0.004	0.13
^C16:0^∆9_index_ ^3^		0.0274 ^a^	0.0265 ^a^	0.0346 ^b^	0.0407 ^b^	0.0010	0.02
^C14:0^∆9_index_ ^4^		0.529 ^b^	0.413 ^a^	0.439 ^a^	0.540 ^b^	0.003	0.03
*^t11^*^C18:1^∆9_index_ ^5^		0.103 ^a^	0.201 ^d^	0.133 ^b^	0.167 ^c^	0.002	0.02
^∑^∆9_index_ ^6^		0.310 ^a^	0.328 ^b^	0.315 ^a^	0.330 ^b^	0.002	0.04
^Σ∆9,6,5,4^FA_index_ ^7^		0.502 ^a^	0.516 ^b^	0.527 ^b^	0.516 ^b^	0.002	0.03

Abbreviations for FA and other items; see Table 3. Means with different superscripts within a row are significantly different at *p* ≤ 0.05. ^1^ The content sum of MUFA (ΣMUFA) = *c7*C16:1 + *c9*C16:1 + *t11*C18:1 + *c6*C18:1 + *c7*C18:1 + *c9*C18:1 + *c11*C18:1 + *c12*C18:1 + *c11*C20:1 + *c11*C22:1 + *c13*C22:1. ^2^ ∆9-desaturaturation of C18:0 index: ^C18:0^∆9_index_ = c9C18:1/(C18:0 + c9C18:1). ^3^ ∆9-desaturaturation of C16:0 index: ^C16:0^∆9_index_ = *c9*C16:1/(C16:0 + *c9*C16:1). ^4^ ∆9-desaturaturation of C14:0 index: ^C14:0^∆9_index_ = *c9*C14:1/(C14:0 + *c9*C14:1). ^5^ ∆9-desaturaturation of *trans*11C18:1 (t11C18:1) index: *^t11^*^C18:1^∆9_index_ = *c9t11*C18:2/(*t11*C18:1 + *c9t11*C18:2). ^6^ Index of ∆9-desaturaturation of C18:0, C16:0, C14:0, and *t11*C18:1: ^∑^∆9_index_ = (*c9*C18:1 + *c9*C16:1 + *c9*C14:1 + *c9t11*C18:2)/(*c9*C18:1 + *c9*C16:1 + *c9*C14:1 + *c9t11*C18:2 + *t11*C18:1 + C14:0 + C18:0 + C16:0). ^7^ Total FA desaturation index (i.e., ∆9-, ∆6-, ∆5-, and ∆4-desaturation of FA): ^Σ∆9,6,5,4^FA_index_ = (ΣMUFA + ΣPUFA)/(C16:0 + C18:0 + C20:0 + C22:0 + C24:0 + ΣMUFA + ΣPUFA).

**Table 5 animals-14-00133-t005:** The contents (µg/g spleen) of selected individual polyunsaturated FA (PUFA), the content ratios of analysed PUFA to ΣFA, indices of elongases and desaturases, and hypocholesterolemic/hypercholesterolemic fatty acid ratio (h/H-Ch ratio) [45] in the ovine spleens.

	Additive: Group/Diet:	-	CA	CA and Se^Ye^	CA and Se^6^	SEM	*p* Value
Item		Control	CA	Se^Ye^CA	Se^6^CA		
*c9t11*CLA		30.3 ^a^	38.9 ^b^	32.1 ^ab^	47.6 ^c^	0.6	0.04
*c9c12*C18:2 (LA)		682 ^c^	550 ^a^	600 ^ab^	645 ^bc^	13	0.03
*c9c12c15*C18:3 (αLNA)		10.6 ^b^	5.5 ^a^	7.2 ^a^	17.0 ^c^	0.5	0.02
*c11c14*C20:2		41.0 ^b^	21.6 ^a^	18.4 ^a^	37.9 ^b^	0.8	0.02
*c8c11c14*C20:3		68.4 ^bc^	49.9 ^a^	61.6 ^ab^	76.0 ^c^	2.9	0.03
*c5c8c11c14*C20:4 (AA)		2 904 ^b^	2 141 ^a^	2 203 ^a^	2 389 ^a^	32	0.05
*c5c8c11c14c17*C20:5 (EPA)		91.4 ^c^	54.4 ^a^	72.7 ^b^	75.9 ^b^	4.1	0.04
*c7c10c13c16c19*C22:5 (DPA)		438	404	443	478	12	0.17
*c4c7c10c13c16c19*C22:5 (DHA)		164 ^a^	135 ^a^	211 ^b^	204 ^b^	8	0.04
Σn-3PUFA ^1^		704 ^b^	599 ^a^	734 ^bc^	774 ^c^	12	0.04
Σn-6PUFA ^2^		3 653 ^c^	2 741 ^a^	2 865 ^ab^	3 111 ^b^	29	0.04
ΣPUFA ^3^		4 428 ^c^	3 400 ^a^	3 649 ^ab^	3 971 ^b^	34	0.03
Σn-6PUFA/Σn-3PUFA		5.189 ^c^	4.576 ^b^	3.903 ^a^	4.018 ^a^	0.005	0.02
Σn-6LPUFA		2 972 ^b^	2191 ^a^	2265 ^a^	2 466 ^ab^	36	0.04
Σn-3LPUFA		693	593	727	757	14	0.27
ΣLPUFA ^4^		3 665 ^b^	2 784 ^a^	2 992 ^a^	3 223 ^ab^	24	0.04
Σn-6LPUFA/Σn-3LPUFA		4.286 ^c^	3.695 ^b^	3.117 ^a^	3.256 ^a^	0.009	0.02
Σn-3LPUFA/ΣFA		0.0359 ^ab^	0.0340 ^a^	0.0395 ^c^	0.0417 ^d^	0.0002	0.04
ΣLPUFA/ΣFA		0.175 ^b^	0.158 ^a^	0.163 ^a^	0.178 ^b^	0.002	0.02
∑PUFA/∑FA		0.215	0.194	0.200	0.218	0.002	0.06
∑PUFA/∑SFA		0.427 ^b^	0.377 ^a^	0.390 ^a^	0.448 ^b^	0.003	0.04
∑UFA/∑SFA		0.938	0.935	0.940	1.045	0.007	0.07
^n−6ElongC20/C18^index ^5^		0.0567 ^c^	0.0378 ^b^	0.0298 ^a^	0.0555 ^c^	0.0002	0.03
^n−3ElongC22/C20^index ^6^		0.721 ^a^	0.888 ^c^	0.862 ^b^	0.864 ^bc^	0.003	0.05
∆4_index_ ^7^		0.272 ^b^	0.250 ^a^	0.323 ^d^	0.297 ^c^	0.001	0.02
∆5_index_ ^8^		0.977	0.977	0.973	0.969	0.002	0.43
h/H-Ch ratio)		2.250 ^b^	2.219 ^a^	2.207 ^a^	2.600 ^c^	0.003	0.02

CLA = conjugated linoleic acid; for other abbreviations for FA and other items, see Table 3 and Table 4. Means with different superscripts within a row are significantly different at *p* ≤ 0.05. ^1^ The content sum of n-3PUFA: Σn-3PUFA = αLNA + *c6c9c12c15*C18:4 + Σn-3LPUFA (i.e., Σn-3LPUFA = *c11c14c17*C20:3 + *c8c11c14c17*C20:3 + EPA + DPA + DHA). ^2^ The content sum of n-6PUFA: Σn-6PUFA = LA + *c6c9c12*C18:3 + Σn-6LPUFA (i.e., Σn-6LPUFA = *c11c14*C20:2 + *c8c11c14*C20:3 + AA + *c7c10c13c16*C22:4). ^3^ The content sum of PUFA: ΣPUFA = ΣCLA + Σn-3PUFA + Σn-6PUFA. ^4^ The content sum of LPUFA: ΣLPUFA = Σn-6LPUFA + Σn-3LPUFA. ^5^ The elongase index of C18:0: n-6ElongC20/C18index = *c11c14*C20:2/(*c11c14*C20:2 + LA). ^6^ The elongase index of C20: ^n−3ElongC22/C20^index = DPA/(DPA + EPA). ^7^ ∆4-desaturation of DPA index = DHA/(DPA + DHA). ^8^ ∆5-desaturation of C20:3n-6 index = AA/(*c8c11c14*C20:3 + AA).

**Table 6 animals-14-00133-t006:** The contents of total cholesterol (TCh; µg/g spleen), tocopherols (µg/g spleen), and MDA (µg/g spleen) ^1^ and values of the modified atherogenic index (_index_A^SFA/ΣToc^) and PUFA peroxidation index (^ΣPUFA^MDA_index_) in the spleen of ewes fed the experimental and control diets.

Item	Group/Diets				SEM	*p* Value
	Control	CA	Se^Ye^CA	Se^6^CA		
TCh	223 ^b^	120 ^a^	308 ^c^	260 ^bc^	23	0.04
δ-tocopherol (δ-T)	1.07	0.33	0.38	0.65	0.05	0.09
γ-tocopherol (γ-T)	0.36	0.23	0.17	0.24	0.04	0.42
α-tocopherol (α-T)	3.83 ^a^	4.65 ^a^	12.11 ^b^	10.93 ^b^	0.06	0.04
α-tocopheryl acetate (α-TAc)	0.11	0.11	0.28	0.28	0.04	0.19
Σ(α-T + α-TAc)	3.93 ^a^	4.75 ^a^	12.39 ^b^	11.22 ^b^	0.07	0.03
Σall-Ts ^2^	5.36 ^a^	5.31 ^a^	12.93 ^b^	12.10 ^b^	0.07	0.03
_index_A^SFA/ΣToc^ [39]	0.0769 ^c^	0.0659 ^b^	0.0258 ^a^	0.0250 ^a^	0.0005	0.03
MDA	4.52	4.45	3.97	3.62	0.12	0.37
^ΣPUFA^MDA_index_ ^3^	1.021 ^b^	1.309 ^d^	1.087 ^c^	0.911 ^a^	0.014	0.03

MDA—malondialdehyde; other abbreviations for FA and other items see Table 3, Table 4 and Table 5. Means with different superscripts within a row are significantly different at *p* ≤ 0.05. ^1^ The content of MDA (C_MDA_) was determined immediately after the spleen homogenization. ^2^ The content sum of all analysed tocopherols: C_Σall-Ts_ = α-Tac + α-T + δ-T + γ-T. ^3 ΣPUFA^MDA_index_ = MDA (ng/g)/ΣPUFA (µg/g).

## Data Availability

Data are contained within the article.

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
