# Peer review of "Organic and Inorganic Selenium Compounds Affected Lipidomic Profile of Spleen of Lambs Fed with Diets Enriched in Carnosic Acid and Fish Oil"

_animals, 2023, doi:10.3390/ani14010133_

Round 1

Reviewer 1 Report

Comments and Suggestions for Authors

This well-done study provides insights into ovine spleen lipid metabolism and the effects of selenium and carnosic acid as antioxidant supplementation. However, it is essential to be more specific on some points.

Author Response

Reviewer #1

Comments and Suggestions for Authors:

  1. What do you mean by underprivileged?

Our response: Lines 18-19: We replaced “the underprivileged consumers.” with “especially poorly nourished consumers.”

  1. Selenium methionine?

Our response: Line 25:  We replaced “selenized yeast (SeYe)“ with “and yeast rich in seleno-methionine (SeYe) …“;

  1. Did this mixture contain selenium?

Our responsOur response: A mineral and vitamin mixture contains sodium selenite (Se4+) – see Table 1 (lines: 187-203);

  1. Although the reader is referred to a previous work with the same diet used in this study, it would be easier to see the ingredients and their composition in this article. Was selenium calculated in the ingredients used?

Our response: Table 1 shows the ingredients of the basal diet (BD) and the chemical profile of supplements. The concentration of Se in the ingredients of the BD (see Table 1), the control and the experimental diets were presented in Table 2 (see footnotes). 

  1. You are discussing the effects on rumen microbiota; however, you did not do any research on that.

Our response: The chemical profiles of fatty acids, tocopherols as well as content of cholesterol in tissues, internal organs (including a spleen) of lambs (ruminants) depended upon rumen bacterial metabolism of the ingredients of diets and supplements (i.e., CA, Se6 and SeYe). However, on the other hand, chemical profiles of additives (like R-O and F-O) and chemical forms of dietary supplements affect the efficiency of bacterial metabolism in the ovine rumen.

We are very grateful Reviewer #1 for the time and effort they spared for revision of our article.

Reviewer 2 Report

Comments and Suggestions for Authors

The manuscript investigates the effects of various selenium sources in feed on the lipid composition of sheep spleens. Overall, the study addresses an interesting topic; however, there are some concerns that need to be addressed for the paper to meet the standards for publication. Particularly, the authors should provide clear details regarding the experimental purpose and significance. Please consider the following points:
1. The introduction section lacks clarity regarding the purpose and significance of the experiment. The literature review suggests uncertainty about the impact of spleens on lipid metabolism in ruminants. However, the study primarily investigates the influence of selenium on spleen lipid composition, this does not effectively address the issues presented in the introduction.
2. The rationale for studying the impact of selenium on spleen lipid composition is not sufficiently justified in the introduction. The relationship between selenium and spleen lipid composition is not clearly established. The author proposes in the introduction that an abundance of polyunsaturated fatty acids can lead to oxidative stress, and selenium has the ability to counteract this stress. Therefore, the author suggests the need to supplement selenium, particularly when feeding diets rich in polyunsaturated fatty acids. However, this still fails to establish a clear connection between selenium and the lipid composition of the spleen, leaving the rationale for conducting this experiment unclear. Additionally, the introduction lacks an explanation for the substantial inclusion of fish oil and carnosic acid in lamb feed. This aspect requires further elucidation in the introduction to explicitly define the purpose and significance of the experiment.
3. The author mentions in both the introduction and conclusion that this experiment can enhance the edible value of the spleen by regulating its lipid composition, thereby reducing the risk of atherosclerosis. The measured indicators in the paper also include "Atherogenic index" and other relevant food science parameters. Firstly, I harbor doubts about the edibility of the spleen. Even if, as the author suggests, the spleen may be considered an inexpensive source of bioactive lipids for individuals at risk of malnutrition,  the idea that the consumption of inadequately composed spleen lipids might lead to atherosclerosis seems unlikely. I recommend the authors to reconsider and clarify the experimental objectives.
4. The discussion section requires improvement in terms of structure and logic to enhance readability. For instance, the first paragraph primarily addresses selenium toxicity, but it includes literature on selenium's impact on rumen microbiota without clarification on its relevance. Furthermore, there is no data related to these two aspects in the present experiment. It is recommended to revise the discussion.
5. The manuscript mentions Se4 and Se6 without providing clear explanations. It only mentions the addition of selenium salts without specifying which salts were used, such as sodium selenite or another form. Additionally, it would be beneficial to include the feed formula for a comprehensive understanding of the experimental conditions.

Author Response

Reviewer #2

Comments and Suggestions for Authors:

[1]. The introduction section lacks clarity regarding the purpose and significance of the experiment. The literature review suggests uncertainty about the impact of spleens on lipid metabolism in ruminants. However, the study primarily investigates the influence of selenium on spleen lipid composition, this does not effectively address the issues presented in the introduction.

[The first part of comments #2]: The rationale for studying the impact of selenium on spleen lipid composition is not sufficiently justified in the introduction. The relationship between selenium and spleen lipid composition is not clearly established. The author proposes in the introduction that an abundance of polyunsaturated fatty acids can lead to oxidative stress, and selenium has the ability to counteract this stress. Therefore, the author suggests the need to supplement selenium, particularly when feeding diets rich in polyunsaturated fatty acids. However, this still fails to establish a clear connection between selenium and the lipid composition of the spleen, leaving the rationale for conducting this experiment unclear.

Our response to [1 ] and [the first part of comment #2]

The effect of dietary Se-compounds (SeYe and Se6) on the lipid composition of the spleen has been extensively explained:

see lines 79-83, 110-126, 147-153, especially 523-538, 543-546 and 551-555.

[The second part of comment # 2]:  Additionally, the introduction lacks an explanation for the substantial inclusion of fish oil (F-O) and carnosic acid (CA) in lamb feed. This aspect requires further elucidation in the introduction to explicitly define the purpose and significance of the experiment.   

Our response

The inclusion of fish oil (F-O) and carnosic acid (CA) in lamb feed has been extensively explained:

  • explanation regarding dietary fish oil (F-O): see lines 101-111, 141-145;
  • explanation regarding dietary carnosic acid (CA): see lines 128-145.

[3].  The author mentions in both the introduction and conclusion that this experiment can enhance the edible value of the spleen by regulating its lipid composition, thereby reducing the risk of atherosclerosis. The measured indicators in the paper also include "Atherogenic index" and other relevant food science parameters. Firstly, I harbor doubts about the edibility of the spleen. Even if, as the author suggests, the spleen may be considered an inexpensive source of bioactive lipids for individuals at risk of malnutrition,the idea that the consumption of inadequately composed spleen lipids might lead to atherosclerosis seems unlikely. I recommend the authors to reconsider and clarify the experimental objectives.

Our response: The experimental objectives have been made more in-depth and expanded:

The reviewer doubts:  “Firstly, I harbor doubts about the edibility of the spleen”- our response (see lines 163-165): “Obviously, examining the impact of SeYe and Se6 added to the ovine diet on the health-promoting properties of spleen is the secondary goal of our current studies.”

The main aims of our study (see lines 155- 163): “Therefore, the principal objectives of our study were to evaluate the effect of SeYe (yeast rich in Se-Met) and Se6 (Na2SeO4) on the lipidomic profile, the concentration of Ts and the yield of lipid peroxidation (i.e. the MDA level) in the ovine spleen. It is of importance not only from the point of view of ruminant physiology and particularly welfare of lambs with diminished splenic functions but also human nutrition, especially in undeveloped countries. Indeed, the spleen, classified as giblets, may be considered as inexpensive source of bioactive lipids for humans at risk of malnutrition. However, animal products containing oxidized lipids, oxidized forms of Chol and highly toxic MDA are harmful to consumers' health.”

Moreover, please refer to  lines 563-574 and the final conclusion (lines 571-574):That is why obtained results seem promising not only when animal welfare, physiology and health are considered (the principal goals of our study), but also when human nutrition is taken into account (the secondary goal of our study).”

[4]. The discussion section requires improvement in terms of structure and logic to enhance readability. For instance, the first paragraph primarily addresses selenium toxicity, but it includes literature on selenium's impact on rumen microbiota without clarification on its relevance. Furthermore, there is no data related to these two aspects in the present experiment. It is recommended to revise the discussion.

Our response: An appropriate improvements in the Discussion section were made: see lines 445-456 and 460-468.

[5].  The manuscript mentions Se4 and Se6 without providing clear explanations. It only mentions the addition of selenium salts without specifying which salts were used, such as sodium selenite or another form. Additionally, it would be beneficial to include the feed formula for a comprehensive understanding of the experimental conditions.

Our response: We have made appropriate improvements in the "Introduction and Materials" and "Methods" sections:

see lines 121-124, 155-156 (we specified selenium-salts: sodium selenite and sodium selenate);

see Table 1 and lines 188-205 and 242-252 (we provided the chemical profile of the basal diet (BD), supplements (CA, SeYe and Se6) and the chemical composition of fish oil (F-O) and rapeseed oil (R-O).

 We are very grateful Reviewer #2 for the time and effort they spared for revision of our article.

Reviewer 3 Report

Comments and Suggestions for Authors

Comments to the manuscript ID 2740483 “Organic and inorganic selenium compounds affected lipidomic profile of spleen of lambs fed with diets enriched in carnosic acid and fish oil”. This study analyzed the effect of organic and inorganic Se supplemented in diets enriched with oil over lambs performance and lipids profile in the animal´s spleen.

Line 27-30 Diets description is a little bit confuse, I suggest to rewrite this par. Also, in the methodology section.

Table1. First, this table presents results and in fat it was located in the Methodology section, so I suggest to do another table, take the results in present them in the results section. Also, could you add the p-values?

In the Methodology, I wonder why authors did not measured lipids profile in blood during the trial (i.e. before (week 0) and after (week 5), or every week).

Line 271-271, So the decrease of C17:0 and C18:0 in the diets except in BD, was due to what?

Author Response

Reviewer #3

  1. Line 27-30 Diets description is a little bit confuse, I suggest to rewrite this par. Also, in the methodology section.

Our response: We would like to kindly ask Reviewer #3 for  using the previous experimental diet descriptions (i.e., CA diet, SeYeCA diet and Se6CA diet) because in our opinion they provide information on supplements added to the diet. Moreover, these descriptions were successfully used in our earlier publications, that is why we will be very grateful if we could also apply them in present manuscript.

  1. First, this table presents results and in fat it was located in the Methodology section, so I suggest to do another table, take the results in present them in the results section. Also, could you add the p-values?

Our response: This Table (with experimental results) was transferred to the Results section (in the revised manuscript this is Table 2).

  1. In the Methodology, I wonder why authors did not measured lipids profile in blood during the trial (i.e. before (week 0) and after (week 5), or every week).

Our response: The lipids profiles in ovine blood were presented in our previous paper: [1] Czauderna, M.; BiaÅ‚ek, M.; Krajewska, K.A.; RuszczyÅ„ska, A.; Bulska, E. Selenium Supplementation into Diets Containing Carnosic Acid, Fish and Rapeseed Oils Affects the Chemical Profile of Whole Blood in Lambs. J Anim Feed Sci 2017, 26, 192–203, doi:10.22358/jafs/76594/2017;

  1. Line 271-271, So the decrease of C17:0 and C18:0 in the diets except in BD, was due to what ?

Our response: We have made the appropriate correction: see lines  338-340:  “Compared to the spleens of the control lambs (the control group), all experimental diets substantially reduced the C17:0 and C18:0 contents in the ovine spleens.”

We are very grateful Reviewer 3 for the time and effort they spared for revision of our article.

Reviewer 4 Report

Comments and Suggestions for Authors

Very interesting study and important results were obtained. 

Nice Work. Congratulations to the research team.

Comments on the Quality of English Language

Please pay attention and rectify  some english grammar, as the plural/singular in some sentences. For instance,  authors often refer "these studies" but the manuscript only presents the result of one study   (eg line 24, 125, 495). Also in line 13 " the spleen, traditionally associated with its role (...) ( not their role. 

Author Response

  1. Please pay attention and rectify  some english grammar, as the plural/singular in some sentences. For instance,  authors often refer "these studies" but the manuscript only presents the result of one study   (eg line 24, 125, 495). Also in line 13 " the spleen, traditionally associated with itsrole (...) ( not their 

Our response:  As suggested by the Reviewer, manuscript was checked by the native speaker and all the corrections indicated by Reviewer #4 have been made.

We are very grateful Reviewer #4 for the time and effort they spared for revision of our article.

Reviewer 5 Report

Comments and Suggestions for Authors

The dosages should be mentioned in the abstract. 

Results are ambiguous in the abstract; there is more likely only a conclusion.

 MM

Are age and BW the only criteria for selecting animals? Then, you used a blocking criteria?

Please mention the origin of the antioxidants.

When you mentioned that each animal consumed 37.8 kg of the diet, is it average data, or was it controlled?

The discussion is adequate and talks about the main findings of this research.

Author Response

Reviewer #5

  1. The dosages should be mentioned in the abstract. 

Our response: We added doses of dietary supplements and oils to the control and experimental diets (see the ABSTRACT; lines 24-44)

  1. Results are ambiguous in the abstract; there is more likely only a conclusion.

Our response:  As suggested, the most important results were added to the ABSTRACT (see the ABSTRACT; lines 24-44)

  1. Are age and BW the only criteria for selecting animals? Then, you used a blocking criteria?

Our response: The body weight and age of Corriedale male lambs were used for selecting of animals (see lines 170-173). No blocking criteria was used for animals selection.

  1. Please mention the origin of the antioxidants.

Our response: We mentioned the origin of the antioxidants: Se6, CA and SeYe – see lines 227-228; 234-235 and 236-239, respectively.

  1. When you mentioned that each animal consumed 37.8 kg of the diet, is it average data, or was it controlled?

Our response: The sheep's diet intake was daily carefully controlled, weigh of both given diet and leftovers was checked. On such basis we claim that on average each lamb ate 37.8 kg of the diet. See lines 210-215.

  1. The discussion is adequate and talks about the main findings of this research.

Our response: We are very grateful for the time and effort they spared for revision of our article.
